# Nutrient Intake during Pregnancy and Adherence to Dietary Recommendations: The Mediterranean PHIME Cohort

**DOI:** 10.3390/nu13051434

**Published:** 2021-04-24

**Authors:** Federica Concina, Paola Pani, Claudia Carletti, Valentina Rosolen, Alessandra Knowles, Maria Parpinel, Luca Ronfani, Marika Mariuz, Liza Vecchi Brumatti, Francesca Valent, D’Anna Little, Oleg Petrović, Igor Prpić, Zdravko Špirić, Aikaterini Sofianou-Katsoulis, Darja Mazej, Janja Snoj Tratnik, Milena Horvat, Fabio Barbone

**Affiliations:** 1Clinical Epidemiology and Public Health Research Unit, Institute for Maternal and Child Health–IRCCS ‘Burlo Garofolo’, via dell’Istria 65/1, 34137 Trieste, Italy; federica.concina@burlo.trieste.it (F.C.); paola.pani@burlo.trieste.it (P.P.); claudiaveronica.carletti@burlo.trieste.it (C.C.); alessandra.knowles@burlo.trieste.it (A.K.); luca.ronfani@burlo.trieste.it (L.R.); 2Department of Medical Area–DAME, University of Udine, via Colugna 50, 33100 Udine, Italy; maria.parpinel@uniud.it (M.P.); marika.mariuz@uniud.it (M.M.); fabio.barbone@uniud.it (F.B.); 3Scientific Direction, Institute for Maternal and Child Health–IRCCS ‘Burlo Garofolo’, via dell’Istria 65/1, 34137 Trieste, Italy; liza.vecchibrumatti@burlo.trieste.it; 4Institute of Hygiene and Clinical Epidemiology, Friuli Centrale Healthcare and University Trust, via Colugna 50, 33100 Udine, Italy; francesca.valent@asufc.sanita.fvg.it; 5Medical Director’s Office, Azienda Sanitaria Friuli Occidentale, via Piave 54, 33170 Pordenone, Italy; dana.little@asfo.sanita.fvg.it; 6Department of Pediatrics, University Hospital Centre Rijeka, Krešimirova 42, 51000 Rijeka, Croatia; oleg@kbc-rijeka.hr (O.P.); igor.prpic@medri.uniri.hr (I.P.); 7Green Infrastructure Ltd., Fallerovo Setaliste 22, 10000 Zagreb, Croatia; zdravko_spiric@hotmail.com; 8Institute of Child Health, “Aghia Sophia” Children’s Hospital, Thivon & Papadiamantopoulou, Goudi, 115 27 Athens, Greece; katsofkat@hotmail.com; 9Jozef Stefan Institute, Jamova 39, SI-1000 Ljubljana, Slovenia; darja.mazej@ijs.si (D.M.); janja.tratnik@ijs.si (J.S.T.); milena.horvat@ijs.si (M.H.)

**Keywords:** prospective cohort study, pregnant women, nutrient intake, dietary reference values, food groups, PHIME

## Abstract

Few studies provide a detailed description of dietary habits during pregnancy, despite the central role of nutrition for the health of the mother and offspring. This paper describes the dietary habits, energy and nutrient intake in pregnant women from four countries belonging to the Mediterranean PHIME cohort (Croatia, Greece, Italy and Slovenia) and evaluates their adherence to the European Food Safety Authority (EFSA) recommendations. A total of 1436 women were included in the present analysis. Maternal diet was assessed using a food frequency questionnaire (FFQ). The mean macro and micronutrient intakes were estimated and compared with the dietary reference values (DRVs). The percentage distribution of the 16 food groups in the total intake of each macronutrient was estimated. All women shared a similar diet during pregnancy; almost all the women in the four countries exceeded the DRV for sugars, and the total fat intake was above the DRV in most women in all the countries, as was the contribution of saturated fatty acids (SFAs) to the total energy intake. In all four countries, we observed an increased risk of micronutrient deficiency for iron, folate and vitamin D. Shared guidelines, implemented at both the national and European level, are essential to improve the maternal nutritional status during pregnancy.

## 1. Introduction

The central role of nutrition in pregnancy for the health and well-being of pregnant women, for pregnancy outcomes and long-term health and for the development of the offspring has been generally recognized and is supported by the most recent scientific literature [1]. In particular, current research indicates that specific maternal conditions prior to and during the gestational period, such as an excessive maternal pre-pregnancy body mass index (pre-pregnancy BMI ≥ 30) and inadequate gestational weight gain (GWG) [2], could affect the immediate and long-term health of the child (e.g., large for gestational age infants, childhood and adult obesity, adult cardiovascular diseases, neurodevelopmental outcomes such as autism spectrum disorder and infant emotional or behavioral problems) and may predispose the mother to complications during pregnancy and delivery (e.g., miscarriage, gestational diabetes, gestational hypertension, preterm birth and pre-eclampsia) [3,4,5]. In addition, an underweight pre-pregnancy status and insufficient GWG during pregnancy might be accompanied by preterm birth, infants small for their gestational age and failure to initiate breastfeeding [6].

The adoption of appropriate dietary behaviors during pregnancy and after childbirth is all the more important if we consider the implications of the maternal diet on the development of food preferences in early life and, by extension, on the child’s lifelong eating habits [7]. Nutrient requirements are considerably increased during the gestational period and stand in contrast to the modest increase in total energy intake recommended for the three trimesters. Women should consume a varied and balanced diet rich in key nutrients rather than eating more [8]. However, scientific evidence indicates that under and overnutrition in women and children are major global health issues, and pregnant women are at increased risk of macro- and micronutrient deficiency, in particular of docosahexaenoic acid (DHA), iron, iodine, calcium, folic acid and vitamin D [8,9].

There are, unfortunately, few studies that provide a detailed description of the dietary intake of pregnant women and address the issue of compliance with dietary recommendations [10,11]. For this reason, the assessment of dietary intake in a large mother–child cohort can be a useful tool for investigating potential diet–disease associations and assisting health authorities and scientific societies when issuing health recommendations that concern this critical and most vulnerable period [12]. In 2007, a Mediterranean prospective mother–child cohort was established as part of the Public Health Impact of Long-Term, Low-Level Mixed Element Exposure in Susceptible Population Strata (PHIME) project. Pregnant women living in four different coastal regions of Italy, Slovenia, Croatia and Greece were enrolled (Mediterranean PHIME cohort), and extensive information on their diets during pregnancy was collected [13].

The main objective of the present paper is to provide a description of the dietary habits, energy and nutrient intake in pregnant women enrolled in the four countries of the Mediterranean PHIME cohort. The secondary objective of the paper is to evaluate the adherence of women enrolled in the cohort to the dietary recommendations proposed by the EFSA [14]. The present study allows for better understanding eating patterns during pregnancy and identifying critical aspects that should be addressed by health care services and health professionals through effective interventions for the promotion of healthy eating habits.

## 2. Materials and Methods

### 2.1. Study Population

A detailed description of the study protocol, with inclusion and exclusion criteria, was published elsewhere [13,15]. The PHIME project was funded by the European Commission’s Sixth Framework Programme for Research and Technological Development, and the main aim of this prospective cohort study was to assess the association between low-level mercury exposure from food consumption during pregnancy and child neurodevelopment at the age of 18 months [13]. The recruitment took place at the Institute for Maternal and Child Health IRCCS Burlo Garofolo in Trieste, Italy (I); at the Maternity Hospital of the University Medical Centre of Ljubljana, Slovenia (S); at the University Hospital of Rijeka, Croatia (C); and at the general regional hospitals of Mytilini (Lesvos), Chios, Samos and Leros in Greece (G). During the recruitment period, eligible women were approached for consent after routine morphologic ultrasound scans between 20 and 22 gestational weeks (I), at routine visits between 34 and 38 gestational weeks (C) or during hospital stay after delivery (S, C and G). The timing of enrollment was subject to country-specific logistic considerations. The study protocol involved the administration of three questionnaires: (1) a short questionnaire to identify any excluding conditions and gather some basic information on the family and lifestyle during pregnancy; (2) a long questionnaire to collect demographic, social and anthropometric data of the mothers and infants and information on the health status, pregnancy and delivery habits and lifestyle and dietary habits; and (3) a supplementary questionnaire to update information on the family and the child.

### 2.2. Maternal Dietary Assessment during Pregnancy and BMI Status

The long questionnaire included a detailed food frequency section, comprising 138 food items that assessed maternal dietary habits during pregnancy (third trimester of gestation). The food frequency section was adapted from an FFQ which had been evaluated for validity [16,17,18] and included data on the consumption of hot beverages, milk and sugar (16 items); bread, cereals and first courses (17 items); main courses (28 items); vegetables and pulses (21 items); fresh and dried fruits (22 items); sweets and other foods (21 items); non-alcoholic beverages (7 items); and alcoholic beverages (6 items). The portion size of each item was indicated in the questionnaire (grams, pieces, cups, plates, glasses and cans). The response categories used to indicate the usual frequency of food consumption over the course of time were never, less than once per month, 1–3 times per month, once per week, 2–4 times per week, 5–6 times per week, once a day, 2–3 times per day and more than 3 times per day. Only for non-alcoholic (7 items) and alcoholic beverages (6 items) was the response category never, less than once per month, 1–3 times per month, once per week, 2–4 times per week, 5–6 times per week, once a day, 2–3 times per day, 4–5 times per day and more than 5 times per day. The response categories for each item were converted into continuous values of intake by assigning to each category a consumption level equal to the median value for that category (for example, 2–4 times/week became 3 times/week) [19].

The mothers’ pre-pregnancy weights (kilograms) and heights (centimeters) were self-reported. The BMI (kg/m^2^) was calculated (weight (kg)/height (m^2^)) and categorized according to the World Health Organization (WHO) definitions as underweight (BMI < 18.5 kg/m^2^), normal weight (18.5 kg/m^2^ ≤ BMI < 25.0 kg/m^2^), overweight (25.0 kg/m^2^ ≤ BMI <30 kg/m^2^) or obese (BMI ≥ 30.0 kg/m^2^). The GWG was calculated by subtracting the pre-pregnancy weight from the self-reported weight at delivery and was categorized as insufficient, appropriate or excessive, based on current Institute Of Medicine (IOM) recommendations [20].

### 2.3. Nutrients

The macro and micronutrient intakes were assessed using the Italian Food Composition Database for Epidemiological Study in Italy-BDA (1998 version) [21], while energy was calculated using the mean quantity intake of each macronutrient and by applying the corresponding energy conversion factor. In this paper, we explore the following macro and micronutrients: total proteins, carbohydrates (available, soluble and fiber), fats (total, saturated, monounsaturated and polyunsaturated fatty acids (SFAs, MUFAs, PUFAs), oleic, linoleic, alpha-linolenic acid, eicosapentaenoic acid (EPA), DHA), minerals (sodium Na, potassium K, calcium Ca, iron Fe, zinc Zn and phosphorus P) and vitamins (B1, B2, B3, B6, B12, C, D and E, expressed as α-tocopherol equivalents and folate). Some nutrients, such as iodine and magnesium, were not considered for this study due to the high percentage of missing data in the food composition database used for the analysis.

Foods and beverages were regrouped into 16 food groups, following the methodology proposed by Talamini et al. (2006) [22], and modified as reported in Appendix A.

### 2.4. Ethics

The study was conducted according to the guidelines laid down in the Helsinki Declaration of 1975 as revised in 1983. The research protocol was approved by the ethics committees of the University of Udine, the Institute for Maternal and Child Health IRCCS Burlo Garofolo, the Clinical Center of Rijeka, the Institute of Child Health of Athens and the National Ethics Committee of the Republic of Slovenia. All aspects of the study, including ethics, were monitored annually by the European Commission. All participating subjects gave their informed consent for inclusion before they were enrolled in the study.

### 2.5. Statistics

Women with children born before 37 weeks of gestation (consistently with previous analyses), with energy intake during pregnancy lower than 1000 Kcal or greater than 4000 Kcal or with missing items (in the food frequency section) over 10% were excluded from the present analysis as outliers. These energy intake cut-offs were defined using the range proposed by Willett for the general population (500–3500 kcal/day) [19], being increased by 500 kcal/day, which represented the energy requirement increase recommended by the Italian DRVs for the third trimester of gestation [23]. The 10% cut point for missing data was based on the methodology adopted for the nutritional assessment in the ALSPAC cohort study [24]. The flow chart of the participants included in the present study is reported in the online Appendix A.

The general characteristics of the mother–child pairs in each country were presented as a frequency and percentage distribution or mean, standard deviation (SD) and median for the categorical and continuous variables, respectively. The mean, SD, median and interquartile range (IQR) were computed for each macro and micronutrient in each country. Differences between countries were assessed by the Kruskall–Wallis test (for continuous variables) and the χ2 test (for categorical variables).

The nutrient intake was compared with the DRV suggested by the EFSA [14] and expressed using different indexes: the adequate intake (AI), reference intake range for macronutrients (RI) and population reference intake (PRI). The percentage of pregnant women with intake below, within or above the DRV was estimated for each macro and micronutrient.

For each country, the percentage distribution of the 16 food groups in the total intake of each macronutrient was estimated with the aim of establishing the main food sources.

Statistical significance for all tests was set at a *p*-value of 0.05. SAS (version 9.4 SAS Institute INC., Cary, NC, USA) was used for the statistical analysis.

## 3. Results

In total, 2189 pregnant women were recruited in the Mediterranean PHIME cohort from 2007 to 2009, and of these, 1730 (79%) filled in the detailed long questionnaire [13], and 1436 (65.6%) were included in the present analysis (Appendix A). Compared with those who were included in the present analysis, the mothers excluded were more likely to hold an “elementary or middle school diploma” (31.1% vs. 20.6%, *p*-value ≤ 0.001), more likely to be underweight (8.2% vs. 5.7%) and normal weight (73.0% vs. 69.4%) and less likely to be overweight (11.6 vs. 17.4, *p*-value = 0.05). The occupational and marital status and age at delivery were similar.

Table 1 shows the characteristics of the studied mother–child pairs from the four countries. The mothers’ ages at delivery, pre-pregnancy BMIs, GWGs, occupational and marital statuses and education levels were significantly different among the four countries.

As for BMI, more than half of the mothers in all four countries started pregnancy with a normal weight (from 65.5% in G to 75% in C), but Slovenian and Greek mothers showed higher overweight and obesity percentages (32.1% and 28.4%, respectively). In all four countries, less than 45% of the mothers achieved adequate GWG at the end of pregnancy; in particular, 46.3% of the Croatian and 41.2% of the Slovenian mothers had excessive GWG. On the other hand, 25% and 26.1% of the Italian and Greek mothers, respectively, had insufficient GWG. The mean age at delivery ranged from 28.8 years for the Greek mothers to 33.1 years for the Italian mothers. A marked difference in occupational status was observed among the four countries; the unemployed mothers were 55.6% in G, 14.6% in I, 10.9% in S and 8.3% in C (*p*-value < 0.01). Additionally, the mothers’ educational levels seemed to differ among the four countries, with the percentage of mothers with a higher educational level ranging from 46.2% in C to 88.8% in S (*p*-value < 0.01).

The distribution of energy intake (kcal) varied among the four countries (*p*-value < 0.01). The mean energy intake was 2141 ± 586.9 kcal/day in Slovenia, 2211 ± 667.2 kcal/day in Greece, 2292 ± 661.1 kcal/day in Croatia and 2310 ± 616.2 kcal/day in Italy.

The distribution of macronutrient intakes and the percentage of adherence to recommendations by country are shown in Table 2.

The average daily intakes of available carbohydrates, fibers, total fats and proteins varied among the four countries. The intake recommendations for available carbohydrates were met by more than 75% of the women in all countries, except for Greece (61.5%), where 35.7% of mothers were below the DRV. On the other hand, almost all the women in the four countries exceeded the DRV for soluble carbohydrates (from 87.5% in C to 97.9% in S).

As shown in Figure 1, the main source of soluble carbohydrates was fruits in all four countries, but the second-most relevant source was added sugars from sweets and desserts, in particular among Italian mothers (35%).

The recommendation for fiber intake (≥25 g/day) was met by almost two-thirds of women involved in the study, except for Italian mothers, 46% of which fell below the DRV. The protein RI (10–20 E%) was met by almost all women (from 95.6% to 97.5%) in all four countries. For Croatian and Slovenian mothers, the main source of protein came from meat (33.9% and 30.3%, respectively), whereas milk and dairy products were the primary source for Italian and Greek mothers (22.4 and 22.3%, respectively). The percentage of proteins derived from pulses and eggs was very low in all four countries (3.9–6.1% pulses; 1.5–2.1% eggs), as was the contribution of seafood to the total protein intake (7.6–7.5% G and C, 6.3–5.8% I and S) (Appendix A). The total fat intake was above the DRV in most women in all the countries. In particular, 72% of Greek mothers exceeded the RI (20–35 E%). Furthermore, the contribution of SFAs to the total energy intake was above the DRV (10 E%) in most women and ranged from 69.1% in C to 74.3% in I (Table 3).

As shown in Figure 2, most of the total fats came from dressings (vegetable or animal oil), the second source being meat for Croatian and Slovenian mothers and milk and dairy products for Italian and Greek mothers.

Taken together, these represent the main sources of SFAs in all countries (Figure 3). The contribution of seafood and nuts to the total fat intake was scarce (2.7–3.8%) for all countries (Figure 2), and the main source of PUFAs was dressing in all four countries (Figure 4).

On the whole, mothers from all four countries adhered to the recommendation not to exceed a caffeine intake of 200 mg/day, equivalent to just over two cups of filter coffee or four cups of tea [25]. The mean intake of coffee was less than one serving per week in the four countries. However, the consumption of alcohol was above the recommended levels in all the countries (77.2%, 77.3%, 68.4% and 67.9% in Croatian, Greek, Italian and Slovenian mothers, respectively), with a mean intake of between 0.8 ± 1.3 and 1.6 ± 2.6 alcohol units per week, equivalent to a mean intake between 1.5 ± 2.4 and 3 ± 4.9 g per day of alcohol in S and G, respectively (data not shown).

Table 4 presents the women’s estimated average daily intake of micronutrients compared to the DRVs.

A high percentage of women from all four countries reported intakes below the DRV for the following micronutrients: calcium, ranging from 39.3% in I to 48.2% in S; iron, ranging from 56.7% in G to 67.5% in I; zinc, ranging from 41.2% in C to 56.5% in S; the vitamin B group, particularly vitamins B1, B2 and B3, ranging from 56.6% in C to 67.7% in I, from 33.1% in C to 51.4% in I and from 52.2% in C to 72.3% in I, respectively; and folate, ranging from 76.4% in G to 86.2% in I. Supplementation of folic acid is recommended before pregnancy and during the first 12 weeks of gestation. In our study, among the four countries, very few mothers adhered to this recommendation before pregnancy (5.7% in G, 19.8% in C, 39% in I and 42.3% in S (Table 1)). During pregnancy, more than 70% of the women of the overall cohort reported taking vitamins or supplements that probably included folic acid (data not shown).

All of the women in the four countries (100%) reported vitamin D intakes below the DRV, ranging from a mean intake of 2.6 µg/day in S to 3.3 µg/day in C. An average daily sodium intake above the DRV (1500 mg/day) was observed in the majority of women from all four countries, ranging from 55.6% in S to 75% in C (mean intake: 1704 mg/day in S; 1880 mg/day in I; 2011 mg/day in C; and 2103 mg/day in G).

## 4. Discussion

An assessment of the dietary intakes of pregnant women in four European countries along the Mediterranean area was carried out on data from the PHIME prospective cohort study. The results show that, overall, women shared similar behavior in terms of compliance to DRVs, with most not adhering to the energy and macronutrient intake recommendations. While it is true that the data were collected over ten years ago and that dietary habits in pregnancy may have changed since then, the main nutritional concerns that emerged from this assessment (i.e., macro- and micronutrient intake and dietary choices) continue to be relevant and reflect those which are still recognized as the most problematic aspects of nutrition in the general population (e.g., excessive SFA, and soluble carbohydrates and sodium intake). Indeed, the mean energy intakes of the women in our study, despite differing slightly between countries, were in line with those reported in Blumfield’s systematic review for the European region [26], as well as in other studies [27]. Considering that the increase in the energy requirement during the third trimester of pregnancy should be 500 kcal/day, the energy intake reported by all the women in the study (calculated for each cohort population using the Schofield equation based on the mean weight of the women) was lower than recommended. These data stand in contrast with the increase in the prevalence of overweight and obesity and of excessive GWG observed in developed countries [28,29], suggesting that, despite not achieving the target energy intake recommended for pregnancy, women are not in energy deficit during gestation. This trend is also evident in our study, where about one out of five women in C and I and about one out of three women in G and S started pregnancy overweight or obese. It has been widely demonstrated that, in most cases, this is due to inadequate eating habits and low physical activity [30], and the situation does not improve during pregnancy. Only about 40% of the women in the four countries achieved adequate GWG at delivery, suggesting that they did not improve their nutrition status during this period. Similarly, our results show that being underweight before pregnancy and insufficient GWG affected between 2.4% and 7.3% and between 11% and 26.1% of women, respectively. These findings could, in part, explain the inadequate caloric intake of about one-fourth of the women included in the present analysis. The majority of the women in the four countries reported protein and available carbohydrate intakes within DRV ranges, as already observed in other studies [8,26,27]. The high fiber intake in these cohorts was in line with that reported by Saunders in Norwegian women [8], probably because women tend to have a high consumption rate of fruits and vegetables during pregnancy. For the majority of the women in the four countries, the intake of total fat and SFAs was above the DRVs and was consistent with data from a meta-analysis involving pregnant women in high income countries [26]. These have been linked to adverse health outcomes in both mothers and children [31,32], as opposed to the well-known health benefits of a PUFA-rich diet [23]. As expected, most of the fats were derived from animal-based products such as meat and milk-dairy products, whereas the contribution of fish was very low, despite its nutritional benefits and the geographical locations of the four countries. Furthermore, the intake of SFAs, mainly deriving from milk dairy products and meat, had similar percentages among the four countries. A significant proportion of fat was also derived from vegetable oils (mainly olive oil), commonly used in the Mediterranean area as dressing for many dishes according to the Italian Food-based Dietary Guidelines [33]. In our study, they represented the main dietary contributors of PUFAs and oleic acid, in line with what was observed in the diets of Spanish pregnant women [11]. Nuts and seeds are also relevant contributors of PUFA intake, but their consumption was very low in all four countries. Similarly, the contribution of omega3-long chain PUFAs was scarce, especially in S and I, where approximately half of the women did not reach the DRVs for EPA and DHA. Consistent with these findings, women from S and I had the lowest contribution of fish and seafood to the total PUFA intake, even though this food group represents the main source of EPA and DHA. Our findings indicate that the diet of the women in the four countries did not adequately cover the requirements of several key micronutrients. Several other micronutrients were found to be barely adequate for approximately half of the women in the four countries. This was the case for calcium, as already observed by Saunders [8] in Norwegian women, as well as for zinc and vitamins B1, B2 and B3, which were below the recommended levels for just under half the women in the four countries. Nevertheless, the median intakes of all these nutrients were very close to the DRVs. Conversely, the low adherence to the recommended intake of vitamin D, folate and iron, observed in more than half of the study population from the four countries, seems to be in line with the results obtained by other authors [8,26,27,34,35] and could potentially have serious consequences for the developing fetus [36,37,38,39]. The intake of vitamin D derived from diet alone was insufficient in all women from all four countries. This is not surprising, as vitamin D is contained in only a few foods and is mostly produced by endogenous synthesis following exposure to ultraviolet radiation. Furthermore, at the time of the study, there were no clearly shared clinical recommendations regarding vitamin D supplementation during pregnancy. The WHO recently issued a recommendation (2020) stating that vitamin D supplements, at a dose of 5 μg per day, should be given only to pregnant women with suspected vitamin D deficiency [40]. Therefore, in order to accurately assess the adequacy of vitamin D’s status during pregnancy, dietary assessment should be combined with direct measurement of the 25-hydroxyvitamin D (25(OH)D) concentration and sun exposure [35,41]. Low folate intake in pregnancy is in line with the inadequate periconceptional folic acid supplementation reported by women in all four countries. At the time of recruitment, national recommendations were available only in three of the four countries involved in the study (Croatia, Italy and Slovenia). While for Italy the guidelines were official, for the other two countries they were not. The guidelines from all three countries recommended folic acid supplementation from one month before conception until 2–3 months of pregnancy [42]. In spite of the recommendation being the same, however, some differences were observed among the countries. Slovenia was the country with the highest adherence (42.3%), and Greece was the one with the lowest adherence (5.7%). As was also reported in other studies [8,27], excessive sodium intake was reported by the majority of the women in all four countries, and since the sodium contained in salt used in food preparation was not considered in the nutritional analysis, it is very likely that our data underestimated the consumption of this micronutrient.

A major strength of this study was the availability of data from a relatively large multinational cohort established in a specific area of the Mediterranean. This allowed for a comprehensive evaluation of the variations in dietary habits in populations that, although similar in many aspects, are exposed to different cultural and environmental factors. In addition, the adoption of a method based on the prospective assessment of the diets of pregnant women eliminated the risk of recall bias.

On the other hand, assessing the diet only once in the third trimester of pregnancy using an FFQ questionnaire introduced the risk of missing dietary changes that may have taken place at different stages of the pregnancy. In spite of its inherent limitations [19], the FFQ remains the most widely used instrument for the assessment of dietary habits, especially in large cohorts, as reported by other studies [8,24]. Our nutritional analysis did not include micronutrient supplements taken by women during pregnancy. While this omission may limit the comprehensiveness of our data, it is in line with the indications provided by international recommendations for a varied and balanced diet with adequate intake of macro and micronutrients during pregnancy, which only consider nutrients derived from the diet. Finally, our nutritional analysis did not consider iodine and magnesium, two micronutrients which are particularly important during pregnancy, because the higher percentage of missing data in the food composition database could lead to underestimations of the daily intake.

Women’s limited knowledge of good nutritional practices, coupled with poor support by health professionals in terms of the nutritional management of pregnancy from before conception and lack of shared guidelines for health care services, plus the additional burden of social inequalities, all contribute to the overall prevalence of malnutrition during pregnancy. The window marking the first 1000 days of pregnancy offers a unique opportunity to engage with women and their families, as it is a time of intense contact with health care services and health professionals [6]. Health professionals should consider adopting a preventive approach that encourages healthy eating behaviors as early as possible in the pregnancy, in line with national recommendations.

## 5. Conclusions

The present paper shows that pregnant women from the four countries of the Mediterranean PHIME cohort study shared a similar diet in terms of nutrient intake and the same behavior in terms of compliance to DRVs; they exceeded the DRV for soluble carbohydrates, total fat and SFAs and had an increased risk of micronutrient deficiency for iron, folate and vitamin D.

The shared and accepted guidelines, along with adequate nutritional counseling and follow-ups during pregnancy implemented at both the national and European levels are, therefore, essential to improve the maternal nutritional status during pregnancy.

## Figures and Tables

**Figure 1 nutrients-13-01434-f001:**
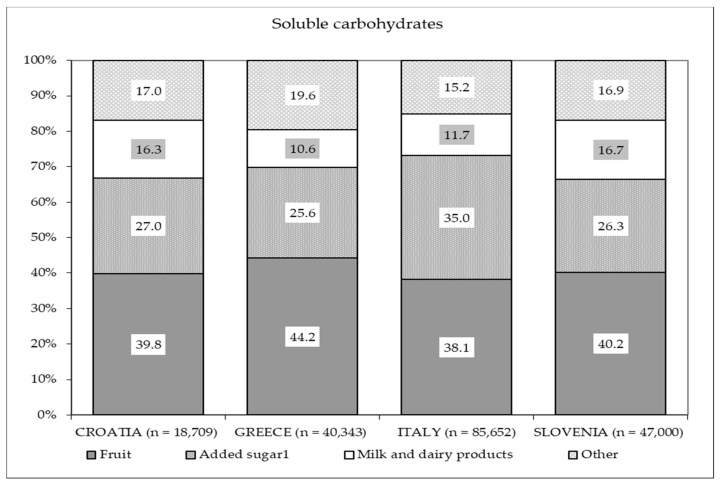
Percentage distribution of the different food sources of soluble carbohydrates based on intake in each country. ^1^ Added sugar derived from the sum of two food groups: “sweets and desserts” and “sugar”.

**Figure 2 nutrients-13-01434-f002:**
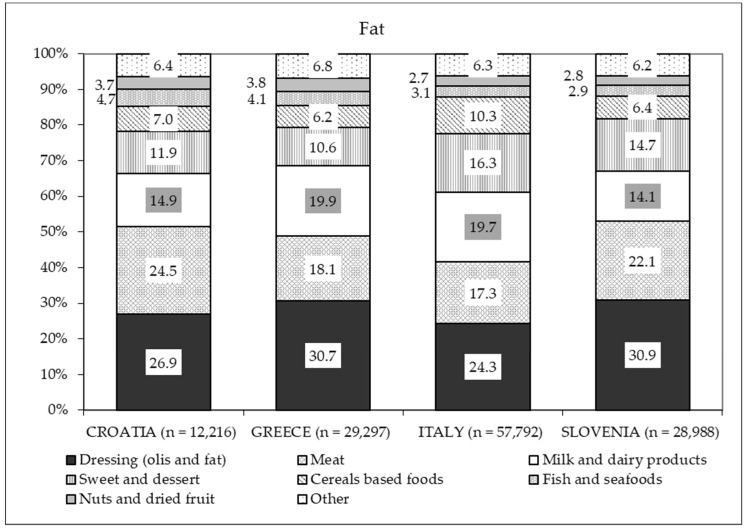
Percentage distribution of the different food sources of fats based on intake in each country.

**Figure 3 nutrients-13-01434-f003:**
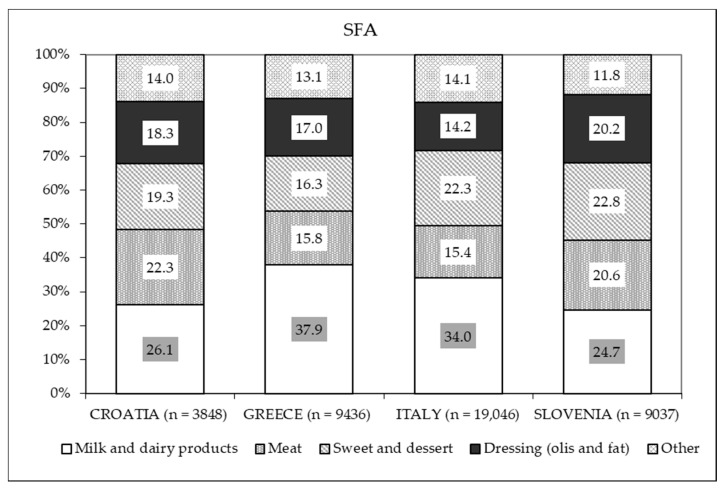
Percentage distribution of the different food sources of saturated fatty acids (SFAs) based on intake in each country.

**Figure 4 nutrients-13-01434-f004:**
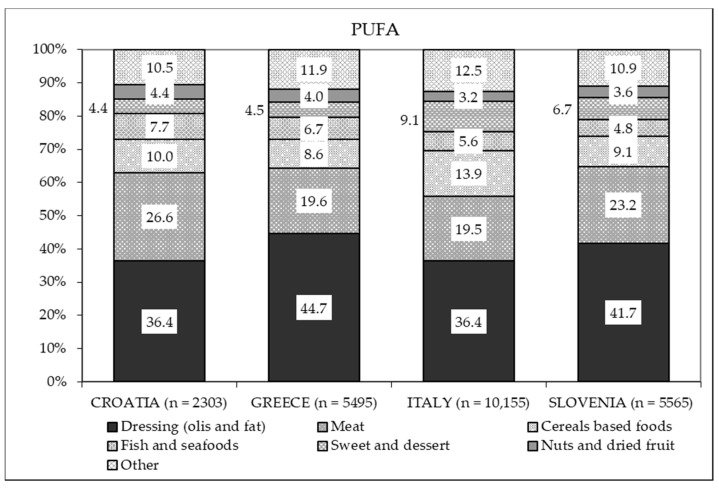
Percentage distribution of the different food sources of polyunsaturated fatty acids (PUFA) based on intake in each country.

**Table 1 nutrients-13-01434-t001:** General characteristics of the mothers and children.

	Croatia (*n* = 136)	Greece (*n* = 314)	Italy (*n* = 646)	Slovenia (*n* = 340)	*p*-Value ^1^
**Mother’s age at delivery, mean ± SD (median)**	30.2 ± 4.4 (30)	28.8 ± 6.3 (29)	33.1 ± 4.3 (33)	30.4 ± 4.3 (30)	<0.01
**Maternal BMI before pregnancy, *n* (%):**					
Underweight	8 (5.9)	19 (6.1)	47 (7.3)	8 (2.4)	<0.01
Normal weight	102 (75.0)	205 (65.5)	466 (72.1)	223 (65.6)	
Overweight	19 (14.0)	63 (20.1)	97 (15.0)	70 (20.6)	
Obese	7 (5.1)	26 (8.3)	36 (5.6)	39 (11.5)	
**Weight gain during pregnancy, *n* (%):**					
Insufficient	15 (11.0)	80 (26.1)	136 (25.0)	74 (22.1)	<0.01
Adequate	58 (42.7)	118 (38.6)	240 (44.1)	123 (36.7)	
Excessive	63 (46.3)	108 (35.3)	168 (30.9)	138 (41.2)	
**Mother’s occupation, *n* (%):**					
Employed	121 (91.7)	139 (44.4)	546 (85.5)	301 (89.1)	<0.01
Unemployed	11 (8.3)	174 (55.6)	93 (14.6)	37 (10.9)	
**Mothers’ marital status, *n* (%):**					
Married or living together	133 (97.8)	311 (99.4)	575 (89.8)	328 (97.0)	<0.01
Widow, single, never married, separated or divorced	3 (2.2)	2 (0.6)	65 (10.2)	10 (3.0)	
**Mother’s educational level, *n* (%):**					
Elementary or middle school	74 (54.8)	75 (24.0)	107 (16.6)	38 (11.2)	<0.01
High school or university degree	61 (45.2)	238 (76.0)	537 (83.4)	300 (88.8)	
**Cigarettes smoked during pregnancy, mean ± SD (median)**	131.9 ± 465.9 (0)	215.2 ± 742.8 (0)	163.3 ± 598.1 (0)	193.6 ± 691.5 (0)	0.39
**Use of vitamins or supplements ^2^ during pregnancy, *n* (%):**					
Yes	106 (81.5)	273 (87.2)	471 (73.8)	282 (83.7)	<0.01
No	24 (18.5)	40 (12.8)	167 (26.8)	55 (16.3)	
**Use of folic acid before pregnancy, *n* (%):**					
Yes	27 (19.8)	18 (5.7)	252 (39.0)	154 (42.3)	
No	109 (80.2)	296 (94.3)	394 (61.0)	186 (54.7)	<0.01
**Child’s gender, *n* (%):**					
Male	61 (44.9)	141 (45.2)	329 (50.9)	165 (48.7)	0.31
Female	75 (55.2)	171 (54.8)	317 (49.1)	174 (51.3)	
**Birth weight (g), mean ± SD (median)**	3549.9 ± 389.9 (3580)	3326.1 ± 429.2 (3300)	3411.5 ± 446.4 (3400)	3455.1 ± 502.9 (3467.5)	<0.01
**Length (cm), mean ± SD (median)**	51.2 ± 2.1 (51.0)	50.2 ± 2.1 (50.0)	50.1 ± 2.0 (50.0)	51.7 ± 3.5 (52.0)	<0.01

^1^ The chi-squared test was applied when variables were categorical, and the Kruskall–Wallis test was applied when variables were continuous. ^2^ Including minerals and herbal products. Abbreviations: SD = standard deviation; *n* = number of subjects; and % = percentage of subjects.

**Table 2 nutrients-13-01434-t002:** Distribution of macronutrient intake in each country.

Macronutrients	Country	Mean	SD	Median	IQR	*p*-Value ^1^	DRV	% of Women below the DRV	% of Women within the DRV	% of Women above the DRV
Available Carbohydrates (g/day)	C	284.4	87.4	281.2	236.9–361.6	<0.01	45–60 E% ^2^	19.1	75.7	5.2
	G	261.9	85.5	251.3	209.2–339.4			35.7	61.5	2.9
	I	292.8	85.5	284.3	244.5–372.4			16.7	77.7	5.6
	S	269.3	80.3	256.8	219.4–350.4			16.5	79.4	4.1
Soluble Carbohydrates (g/day)	C	137.6	57.3	134.6	98.6–176.8	0.11	15 E% ^3^	12.5	-	87.5
	G	128.5	54.2	127.7	88.3–166.8			13.7	-	86.3
	I	132.6	52.0	125.1	99.2–165.2			10.2	-	89.8
	S	138.2	51.5	128.9	105.4–180.8			2.1	-	97.9
Fiber (g/day)	C	29.2	11.3	28.2	23.0–34.4	<0.01	25 g/day ^4^	35.3	-	64.7
	G	30.9	11.7	29.7	23.2–41.6			32.2	-	67.8
	I	27.7	10.9	26.2	21.5–36.6			46.0	-	54.0
	S	29.8	11.2	28.3	23.4–39.2			38.8	-	61.2
Total Fat (g/day)	C	89.8	29.8	84.0	72.9–118.2	<0.01	20–35 E% ^2^	0	48.5	51.5
	G	93.3	31.9	92.1	73.9–119.6			0	28.0	72.0
	I	89.5	27.1	86.0	73.7–114.7			0	52.2	47.8
	S	85.3	25.8	81.6	71.3–109.3			0.6	41.5	57.9
Protein (g/day)	C	91.5	29.2	88.1	74.4–117.3	<0.01	10–20 E% ^2^	0	95.6	4.4
	G	83.9	26.7	81.7	65.1–105.0			0	95.9	4.1
	I	87.1	24.8	85.0	72.8–109.8			0.3	97.5	2.2
	S	80.5	25.9	77.4	64.2–101.8			0.9	96.8	2.4

^1^ The Kruskall–Wallis test was applied. ^2^ Below DRV: the extreme value of the range not included (<); within DRV: the extreme values included; above DRV: the extreme values not included (>). ^3^ Below DRV: extreme value included (≤); above DRV: extreme value not included (>). ^4^ Below DRV: extreme value not included (<); above DRV: extreme value included (≥). Abbreviations: SD = standard deviation; IQR = interquartile range; DVR = dietary reference value; C = Croatia (*n* = 136); G = Greece (*n* = 314); I = Italy (*n* = 646); and S = Slovenia (*n* = 340).

**Table 3 nutrients-13-01434-t003:** Distribution of fat intake in each country.

Fat	Country	Mean	SD	Median	IQR	*p*-Value ^1^	DRV	% of Women below the DRV	% of Women within the DRV	% of Women above the DRV
MUFA (g/day)	C	36.8	12.4	34.2	30.2–49.2	<0.01	10–15 E% ^2^	2.2	61.0	36.8
	G	38.7	12.9	38.7	30.4–50.4			1.3	36.3	62.4
	I	36.8	11.3	35.5	30.5–47.5			1.1	63.5	35.5
	S	35.3	11.0	34.1	29.9–45.1			1.2	53.8	45.0
PUFA (g/day)	C	16.9	5.6	16.5	13.8–22.6	<0.01	5–10 E% ^2^	4.4	94.9	0.7
	G	17.5	5.7	17.5	14.0–23.9			3.5	93.6	2.9
	I	15.7	5.3	15.1	12.7–20.3			17.0	82.2	0.8
	S	16.4	5.4	15.8	13.7–20.8			6.2	92.7	1.2
Linoleic Acid (g/day)	C	13.7	4.6	13.6	11.1–18.0	<0.01	4 E% ^3^	4.4	-	95.6
	G	14.5	4.8	14.4	11.9–19.8			3.2	-	96.8
	I	13.0	4.5	12.4	10.5–16.8			15.0	-	85.0
	S	13.6	4.6	13.1	11.4–17.4			5.3	-	94.7
α-linoleic Acid (g/day)	C	1.8	0.6	1.8	1.5–2.4	<0.01	0.5 E% ^3^	2.2	-	97.8
	G	1.9	0.6	1.9	1.5–2.5			1.0	-	99.0
	I	1.7	0.6	1.6	1.4–2.2			7.9	-	92.1
	S	1.8	0.6	1.7	1.5–2.2			2.4	-	97.4
SFA (g/day)	C	28.3	10.4	26.0	22.5–38.5	<0.01	10 E% ^3^	30.9	-	69.1
	G	30.1	13.2	27.6	21.4–40.7			29.6	-	70.4
	I	29.5	10.0	27.9	24.3–37.4			25.7	-	74.3
	S	26.6	9.0	25.0	21.6–34.0			28.5	-	71.5
EPA + DHA (mg/day)	C	690.8	439.0	569.0	393.5–828.8	<0.01	350–450 mg/day ^2^	17.7	16.9	64.4
	G	571.4	381.6	489.9	314.6–742.7			29.9	15.6	54.5
	I	404.6	264.1	337.7	231.9–489.4			52.8	18.4	28.8
	S	415.6	256.2	365.9	248.8–520.2			45.3	22.4	32.4

^1^ The Kruskall–Wallis test was applied. ^2^ Below DRV: extreme value of the range not included (<); within DRV: extreme values included; above DRV: extreme value not included (>). ^3^ Below DRV: extreme value not included (<); above DRV: extreme value included (≥). Abbreviations: SD = standard deviation; IQR = interquartile range; MUFA = monounsaturated fatty acid; C = Croatia (*n* = 136); G = Greece (*n* = 314); I = Italy (*n* = 646); S = Slovenia (*n* = 340); PUFA = polyunsaturated fatty acid; SFA = saturated fatty acid; EPA = eicosapentaenoic acid; and DHA = docosahexaenoic acid.

**Table 4 nutrients-13-01434-t004:** Distribution of micronutrient intake in each country.

Micronutrients	Country	Mean	SD	Median	IQR	*p*-Value ^1^	DRV	% of Women below (<) DRV	% of Women above (≥) DRV
Calcium (mg/day)	C	1123.9	459.9	1102.1	820.3–1460.1	0.10	1000 mg/day	41.2	58.8
	G	1132.3	496.6	1111.0	791.0–1450.8			43.3	56.7
	I	1142.5	390.9	1100.9	930.5–1432.0			39.3	60.7
	S	1088.4	409.7	1026.1	842.2–1395.3			48.2	51.8
Iron (mg/day)	C	15.3	4.8	14.9	12.1–20.0	<0.01	16 mg/day	58.1	41.9
	G	15.2	5.1	15.1	11.6–19.7			56.7	43.3
	I	14.4	4.6	13.8	11.7–18.2			67.5	32.5
	S	15.2	4.9	14.7	12.4–19.2			60.3	39.7
Phosphorus (mg/day)	C	1601.3	520.6	1550.9	1248.6–2048.7	0.05	550 mg/day	0	100.0
	G	1496.8	507.5	1469.3	1165.2–1873.2			1.0	99.0
	I	1534.1	440.3	1513.2	1281.6–1932.4			0.2	99.8
	S	1479.2	464.7	1417.1	1172.0–1877.8			0.6	99.4
Potassium (mg/day)	C	4855.1	1629.4	4793.6	3910.3–6378.5	<0.01	3500 mg/day	22.8	77.2
	G	4730.9	1605.6	4767.0	3524.4–6138.7			26.4	73.6
	I	4204.3	1454.8	4047.2	3397.8–5330.1			35.5	64.6
	S	4577.0	1448.4	4356.2	3677.2–5811.3			24.1	75.9
Sodium (mg/day)	C	2010.8	682.1	1959.0	1582.7–2509.2	<0.01	1500 mg/day	25.0	75.0
	G	2103.1	899.3	2009.3	1537.0–2667.0			26.1	73.9
	I	1880.3	646.7	1793.8	1466.4–2383.0			32.4	67.7
	S	1704.0	660.2	1571.2	1277.9–2144.8			44.4	55.6
Zinc (mg/day)	C	12.0	3.8	11.5	9.6–15.5	<0.01	11 mg/day	41.2	58.8
	G	11.2	3.5	11.3	8.8–14.2			45.2	54.8
	I	11.5	3.3	11.3	9.7–14.7			47.4	52.6
	S	10.7	3.2	10.3	8.8–16.7			56.5	43.5
Vitamin B1 (mg/day)	C	1.34	0.42	1.30	1.1–1.6	0.10	1.4 mg/day	56.6	43.4
	G	1.27	0.40	1.28	1.0–1.6			60.5	39.5
	I	1.25	0.39	1.21	1.0–1.5			67.7	32.4
	S	1.28	0.38	1.26	1.0–1.5			65.3	34.7
Vitamin B12 (μg/day)	C	8.9	6.1	7.0	5.4–10.7	<0.01	4.5 μg/day	13.2	86.8
	G	8.1	4.9	6.9	4.8–10.3			23.3	76.8
	I	6.1	3.5	5.3	4.2–7.5			35.8	64.2
	S	5.9	4.3	5.1	3.6–7.3			42.7	57.4
Folate (µg/day)	C	465.5	171.3	444.1	363.1–578.3	<0.01	600 µg/day	77.9	22.1
	G	468.8	181.0	460.2	343.8–632.1			76.4	23.6
	I	423.4	161.5	406.3	323.0–542.8			86.2	13.8
	S	441.1	166.0	414.8	341.8–551.7			85.0	15.0
Vitamin B2 (mg/day)	C	2.24	0.81	2.18	1.6–2.8	<0.01	1.9 mg/day	33.1	66.9
	G	2.00	0.73	1.97	1.5–2.5			44.9	55.1
	I	1.97	0.63	1.88	1.5–2.4			51.4	48.6
	S	2.14	0.73	2.03	1.6–2.6			44.4	55.6
Vitamin B3 (mg/day)	C	22.5	7.1	21.6	17.7–28.5	<0.01	22 mg/day	52.2	47.8
	G	20.1	6.2	20.4	16.1–25.2			59.9	40.1
	I	19.0	5.9	18.3	15.2–23.8			72.3	27.7
	S	20.1	6.3	19.8	16.5–25.0			64.7	35.3
Vitamin B6 (mg/day)	C	2.90	0.94	2.79	2.2–3.5	<0.01	1.8 mg/day	11.8	88.2
	G	2.76	0.89	2.78	2.0–3.4			17.2	82.8
	I	2.45	0.79	2.36	1.9–2.9			22.0	78.0
	S	2.74	0.86	2.67	2.1–3.2			13.8	86.2
Vitamin C (mg/day)	C	281.0	146.1	254.5	187.6–362.4	<0.01	105 mg/day	5.9	94.1
	G	308.7	157.2	299.1	187.9–419.8			7.3	92.7
	I	249.7	131.6	220.2	161.6–319.6			8.2	91.8
	S	270.2	140.1	237.1	175.0–345.2			3.8	96.2
Vitamin D (µg/day)	C	3.3	1.6	2.9	2.1–4.2	<0.01	15 µg/day	100.0	0
	G	3.1	1.6	2.7	2.0–4.1			100.0	0
	I	3.1	1.4	2.8	2.1–3.8			100.0	0
	S	2.6	1.3	2.4	1.8–3.1			100.0	0
Vitamin E (mg/day)	C	13.9	5.4	13.5	9.8–16.8	<0.01	11 mg/day	32.4	67.7
	G	15.0	5.4	14.8	10.7–19.2			27.4	72.6
	I	13.5	4.9	12.9	10.0–16.2			33.0	67.0
	S	13.5	4.9	12.8	10.0–16.1			32.9	67.1

^1^ The Kruskall–Wallis test was applied. Abbreviations: SD = standard deviation; IQR = interquartile range; DRV = dietary reference value; C = Croatia (*n* = 136); G = Greece (*n* = 314); I = Italy (*n* = 646); and S = Slovenia (*n* = 340).

## Data Availability

The data described in the manuscript, in the code book and in the analytical code will not be made available because we do not have an accessible repository in which to deposit them. Furthermore, the data belong to each country of the Mediterranean PHIME cohort.

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
