# Peer review of "Nutrient Intake during Pregnancy and Adherence to Dietary Recommendations: The Mediterranean PHIME Cohort"

_nutrients, 2021, doi:10.3390/nu13051434_

Round 1

Reviewer 1 Report

This paper describes dietary habits, energy and nutrients intake in pregnant women from four countries (Croatia; Greece; Italy; Slovenia). The data were collected in the PHIME study, a newborn cohort study carried out in Italy between 2007 and 2010. In the end, the authors used data from this old database and compared the results from four states.

First of all, the 20 authors (!) should clarify that the data are more than 10 years old. The comparison between the 4 countries did not lead to significant results. For this reason, the authors only included considerations of the entire study sample in the discussion.

Author Response

REVIEW REPORT (ROUND 1) – REVIEWER 1

This paper describes dietary habits, energy and nutrients intake in pregnant women from four countries (Croatia; Greece; Italy; Slovenia). The data were collected in the PHIME study, a newborn cohort study carried out in Italy between 2007 and 2010. In the end, the authors used data from this old database and compared the results from four states.

  • First of all, the 20 authors (!) should clarify that the data are more than 10 years old.

The main aim of the Mediterranean prospective PHIME cohort study was to assess the association between low-level mercury exposure from food consumption in pregnancy and child neurodevelopmental at 18 months of age. Thus, our earlier efforts focused specifically on investigating this hypothesis and not on the nutritional aspects of the diet of pregnant women, although the FFQ was designed also with this purpose in mind (Miklavčič A et al., Environ Res, 2013; Valent F et al., J Epidemiol, 2013; Llop S et al., Environ Int, 2017; Barbone F et al., Int J Hyg Environ Health, 2019; Nišević JR et al., Environ Res, 2019; Castriotta L et al., Int J Hyg Environ Health, 2020). We believe that the information provided by these data offers an interesting and specific insight into the dietary behaviour of pregnant women who live in a geographic area which is commonly associated to the Mediterranean diet, but still retains notable cultural differences. Furthermore, the main nutritional concerns, shared by all four countries in the Mediterranean PHIME cohort (i.e., macro- and micro-nutrients intake and dietary choices), reflect those which are still recognized as the most problematic aspects of nutrition in the general population (excessive SFA, soluble carbohydrates and sodium intake, etc.). We modified the discussion section of the paper to better explain this aspect (LINES 309-320).

  • The comparison between the 4 countries did not lead to significant results. For this reason, the authors only included considerations of the entire study sample in the discussion.

In the discussion section, the results from the four countries are presented separately (and treated as distinct groups for the purpose of comparisons). However, we realise that the wording of some of the sentences may be ambiguous in this respect. We modified the discussion to correct this. We substituted the words “study” and “cohort” with “four countries” (LINES 337-338; 342 and 360-361) and the word “this” with “these” (LINE 339).

Reviewer 2 Report

An interesting study which provides a description of dietary habits, energy and nutrients intake in pregnant women enrolled in the four countries of the Mediterranean PHIME cohort.

Author Response

REVIEW REPORT (ROUND 1) – REVIEWER 2

Thank you for giving me the opportunity for review the manuscript entitled “Nutrient intake during pregnancy and adherence to dietary recommendations: the Mediterranean PHIME cohort”.

The manuscript is in scope of the Journal however it requires some clarifications.

The topic is of interest, as pregnant women have special dietary requirements that need to be met to prevent damage to the growing fetus as well as to prevent the development of certain diseases later in life.

Please find the specific comments below:

  • In introduction part of the manuscript, some health consequences for mother, fetus and child related to maternal diet during pregnancy has been provided, however it would be also good to mention neurodevelopmental outcomes as long-term consequences on antenatal diet.

We included this information in the introduction section of the paper, as suggested (LINES 53-54).

  • In introduction part of the manuscript too detailed information is provided for PHIME cohort – this part can be moved into the methods section of the paper (in introduction the cohort can be only mentioned as the base for proposed analyses).

We moved this part into the materials and methods section of the paper, as suggested (LINES 96-100).

  • One of the aim/advantages of this study proposed by authors, namely: “the present study allows to better understand how eating patterns change during pregnancy …” (line 88) cannot be achieved by this project as in this study there is no data (and analyses) before and during pregnancy so the changes cannot be identified. This statement should be removed or modified.

We modified the statement, as suggested (LINES 89).

  • The PHIME cohort was established in 2007 so the authors should comment if the presented results are still relevant (it can be suspected that the dietary habits change over the last 10 years).

The main aim of the Mediterranean prospective PHIME cohort study was to assess the association between low-level mercury exposure from food consumption in pregnancy and child neurodevelopmental at 18 months of age. Thus, our earlier efforts focused specifically on investigating this hypothesis and not on the nutritional aspects of the diet of pregnant women, although the FFQ was designed also with this purpose in mind (Miklavčič A et al., Environ Res, 2013; Valent F et al., J Epidemiol, 2013; Llop S et al., Environ Int, 2017; Barbone F et al., Int J Hyg Environ Health, 2019; Nišević JR et al., Environ Res, 2019; Castriotta L et al., Int J Hyg Environ Health, 2020). We believe that the information provided by these data offers an interesting and specific insight into the dietary behaviour of pregnant women who live in a geographic area which is commonly associated to the Mediterranean diet, but still retains notable cultural differences. Furthermore, the main nutritional concerns, shared by all four countries in the Mediterranean PHIME cohort (i.e., macro- and micro-nutrients intake and dietary choices), reflect those which are still recognized as the most problematic aspects of nutrition in the general population (excessive SFA, soluble carbohydrates and sodium intake, etc.). We modified the discussion section of the paper to better explain this aspect (LINES 309-320).

  • The sentence in line 103-106 is not relevant to the analyses and the results presented in current paper so it should be deleted.

We deleted the non-relevant part of the sentence, as suggested (LINES 108-113)

  • The sentence in line 149-152 should be modified to eliminate the interpretation. I propose to change it as follow: “Some nutrients, such as iodine and magnesium, were not considered for this study due to the high percentage of missing data in the food composition database used for the analysis.” and leave it in methods and additionally add the comment to it into the discussion part of the paper.

We modified the sentence, as suggested (LINES 156-157).

We added a comment on this in the discussion section of the paper, as suggested (LINES 410-413).

  • The methods part of the manuscript should be improved by adding more information (description) about socio-demographic variables. We added information on the socio-demographic variables, as suggested (LINES 115-116).

number of participants included as well as the response rate (the authors mentioned the number of participants only in the results part of the paper – whereas it is important for correct understanding of the methodology of the study). We included the sentence in the materials and methods section of the paper, as suggested (LINES 183-184), and information on the response rate has been added in the results section of the paper (LINES 202-204).

the approval from ethical committee and inform consent. We added this information in the materials and methods section of the paper, as suggested (LINES 164-172).

There is also no information provided about the supplement use (when and how that information was collected).

Data on the use of supplements/vitamins during pregnancy was collected using the long questionnaire filled in by mothers during pregnancy. However, the information collected was limited to whether or not these products were used, as reported in Table 1. This issue is already addressed in the discussion section of the paper (LINES 405-406).

  • It needs to be underlined that diet is not the main source of vitamin D – although it is mentioned in the discussion the authors omitted the recent WHO recommendations: “WHO antenatal care recommendations for a positive pregnancy experience. Nutritional interventions update: Vitamin D supplements during pregnancy. Geneva: World Health Organization; 2020.”. What is more the monitoring of 25(OH)D blood levels should be discussed.

We modified the discussion section of the paper, as suggested (LINES 375-381), making specific reference to the recent WHO recommendation document (LINES 566-569).

  • The strong limitation of this study is that the authors collected the information about supplements/vitamins during pregnancy but have not included such data (for specific micronutrients and vitamins) in the analyses.

We agree that this constitutes a limitation of the study. However, as stated in discussion section of the paper, our method is in line with that of the international recommendations for a varied and balanced diet with adequate intake of macro and micronutrients, which only consider nutrients derived from the diet (LINES 405-410).

  • It should be also more strongly underlined that over or underreporting of nutrients intake could occur when the data are collected by the questionnaire and that the declared food intake may also not correspond to the nutritional status measured by laboratory means (the concentration in blood)  as a result of different bioavailability of nutrients from different food products and individual differences in metabolism. It would be useful (if available) to look at the micronutrients/vitamins level in biological samples (looking at the previous papers from PHIME cohort Zn concentration was assessed in this study). This should be at least discussed in the discussion section of the paper.

We agree with the referee on the limitations of the FFQ which is why we refer to Willett’s publication for a description of its inherent weaknesses (Willett W, 2012)

The suggestion of looking at micronutrients/vitamins levels in biological samples is very interesting, however, the main objective of the Mediterranean prospective PHIME cohort study was to assess the association between low-level mercury exposure from food consumption during pregnancy and child neurodevelopment at 18 months of age. The analysis of other minerals and vitamins in biological samples was not included in the protocol. Only Zn and Se concentrations in maternal blood samples were analysed but only for two of the four participating countries (Italy and Croatia) .

  • I suggest adding the additional section – “conclusion”

We added a conclusion section in the paper (LINES 430-438) by using some parts of the discussion section (LINES 425-428), as suggested.

The manuscript has been checked by a native English-speaking colleague.

Reviewer 3 Report

Thank you for giving me the opportunity for review the manuscript entitled “Nutrient intake during pregnancy and adherence to dietary recommendations: the Mediterranean PHIME cohort”.

The manuscript is in scope of the Journal however it requires some clarifications.

The topic is of interest, as pregnant women have special dietary requirements that need to be met to prevent damage to the growing fetus as well as to prevent the development of certain diseases later in life.

Please find the specific comments below:

  • In introduction part of the manuscript, some health consequences for mother, fetus and child related to maternal diet during pregnancy has been provided, however it would be also good to mention neurodevelopmental outcomes as long-term consequences on antenatal diet.
  • In introduction part of the manuscript too detailed information is provided for PHIME cohort – this part can be moved into the methods section of the paper (in introduction the cohort can be only mentioned as the base for proposed analyses).
  • One of the aim/advantage of this study proposed by authors, namely: “the present study allows to better understand how eating patterns change during pregnancy …” (line 88) cannot be achieved by this project as in this study there is no data (and analyses) before and during pregnancy so the changes cannot be identified. This statement should be removed or modified.
  • The PHIME cohort was established in 2007 so the authors should comment if the presented results are still relevant (it can be suspected that the dietary habits change over the last 10 years).
  • The sentence in line 103-106 is not relevant to the analyses and the results presented in current paper so it should be deleted.
  • The sentence in line 149-152 should be modify to eliminate the interpretation. I propose to change it as follow: “Some nutrients, such as iodine and magnesium, were not considered for this study due to the high percentage of missing data in the food composition database used for the analysis.” and leave it in methods and additionally add the comment to it into the discussion part of the paper.  
  • The methods part of the manuscript should be improved by adding more information (description) about socio-demographic variables, number of participants included as well as the response rate (the authors mentioned the number of participants only in the results part of the paper – whereas it is important for correct understanding of the methodology of the study), the approval from ethical committee and inform consent. There is also no information provided about the supplement use (when and how that information was collected).
  • It needs to be underlined that diet is not the main source of vitamin D – although it is mentioned in the discussion the authors omitted the recent WHO recommendations: “WHO antenatal care recommendations for a positive pregnancy experience. Nutritional interventions update: Vitamin D supplements during pregnancy. Geneva: World Health Organization; 2020.”. What is more the, monitoring of 25(OH)D blood levels should be discussed.
  • The strong limitation of this study is that the authors collected the information about supplements/vitamins during pregnancy but have not included such data (for specific micronutrients and vitamins) in the analyses.
  • It should be also more strongly underlined that over or underreporting of nutrients intake could occur when the data are collected by the questionnaire  and that the declared food intake may also not correspond to the nutritional status measured by laboratory means (the concentration in blood)  as a result of different bioavailability of nutrients from different food products and individual differences in metabolism. It would be useful (if available) to look at the micronutrients/vitamins level in biological samples (looking at the previous papers from PHIME cohort Zn concentration was assessed in this study). This should be at least discussed in the discussion section of the paper.
  • I suggest to add the additional section – “conclusion”

Author Response

REVIEW REPORT (ROUND 1) – REVIEWER 2

Thank you for giving me the opportunity for review the manuscript entitled “Nutrient intake during pregnancy and adherence to dietary recommendations: the Mediterranean PHIME cohort”.

The manuscript is in scope of the Journal however it requires some clarifications.

The topic is of interest, as pregnant women have special dietary requirements that need to be met to prevent damage to the growing fetus as well as to prevent the development of certain diseases later in life.

Please find the specific comments below:

  • In introduction part of the manuscript, some health consequences for mother, fetus and child related to maternal diet during pregnancy has been provided, however it would be also good to mention neurodevelopmental outcomes as long-term consequences on antenatal diet.

We included this information in the introduction section of the paper, as suggested (LINES 53-54).

  • In introduction part of the manuscript too detailed information is provided for PHIME cohort – this part can be moved into the methods section of the paper (in introduction the cohort can be only mentioned as the base for proposed analyses).

We moved this part into the materials and methods section of the paper, as suggested (LINES 96-100).

  • One of the aim/advantages of this study proposed by authors, namely: “the present study allows to better understand how eating patterns change during pregnancy …” (line 88) cannot be achieved by this project as in this study there is no data (and analyses) before and during pregnancy so the changes cannot be identified. This statement should be removed or modified.

We modified the statement, as suggested (LINES 89).

  • The PHIME cohort was established in 2007 so the authors should comment if the presented results are still relevant (it can be suspected that the dietary habits change over the last 10 years).

The main aim of the Mediterranean prospective PHIME cohort study was to assess the association between low-level mercury exposure from food consumption in pregnancy and child neurodevelopmental at 18 months of age. Thus, our earlier efforts focused specifically on investigating this hypothesis and not on the nutritional aspects of the diet of pregnant women, although the FFQ was designed also with this purpose in mind (Miklavčič A et al., Environ Res, 2013; Valent F et al., J Epidemiol, 2013; Llop S et al., Environ Int, 2017; Barbone F et al., Int J Hyg Environ Health, 2019; Nišević JR et al., Environ Res, 2019; Castriotta L et al., Int J Hyg Environ Health, 2020). We believe that the information provided by these data offers an interesting and specific insight into the dietary behaviour of pregnant women who live in a geographic area which is commonly associated to the Mediterranean diet, but still retains notable cultural differences. Furthermore, the main nutritional concerns, shared by all four countries in the Mediterranean PHIME cohort (i.e., macro- and micro-nutrients intake and dietary choices), reflect those which are still recognized as the most problematic aspects of nutrition in the general population (excessive SFA, soluble carbohydrates and sodium intake, etc.). We modified the discussion section of the paper to better explain this aspect (LINES 309-320).

  • The sentence in line 103-106 is not relevant to the analyses and the results presented in current paper so it should be deleted.

We deleted the non-relevant part of the sentence, as suggested (LINES 108-113)

  • The sentence in line 149-152 should be modified to eliminate the interpretation. I propose to change it as follow: “Some nutrients, such as iodine and magnesium, were not considered for this study due to the high percentage of missing data in the food composition database used for the analysis.” and leave it in methods and additionally add the comment to it into the discussion part of the paper.

We modified the sentence, as suggested (LINES 156-157).

We added a comment on this in the discussion section of the paper, as suggested (LINES 410-413).

  • The methods part of the manuscript should be improved by adding more information (description) about socio-demographic variables. We added information on the socio-demographic variables, as suggested (LINES 115-116).

number of participants included as well as the response rate (the authors mentioned the number of participants only in the results part of the paper – whereas it is important for correct understanding of the methodology of the study). We included the sentence in the materials and methods section of the paper, as suggested (LINES 183-184), and information on the response rate has been added in the results section of the paper (LINES 202-204).

the approval from ethical committee and inform consent. We added this information in the materials and methods section of the paper, as suggested (LINES 164-172).

There is also no information provided about the supplement use (when and how that information was collected).

Data on the use of supplements/vitamins during pregnancy was collected using the long questionnaire filled in by mothers during pregnancy. However, the information collected was limited to whether or not these products were used, as reported in Table 1. This issue is already addressed in the discussion section of the paper (LINES 405-406).                                                                                                                                                       

  • It needs to be underlined that diet is not the main source of vitamin D – although it is mentioned in the discussion the authors omitted the recent WHO recommendations: “WHO antenatal care recommendations for a positive pregnancy experience. Nutritional interventions update: Vitamin D supplements during pregnancy. Geneva: World Health Organization; 2020.”. What is more the monitoring of 25(OH)D blood levels should be discussed.

We modified the discussion section of the paper, as suggested (LINES 375-381), making specific reference to the recent WHO recommendation document (LINES 566-569).

  • The strong limitation of this study is that the authors collected the information about supplements/vitamins during pregnancy but have not included such data (for specific micronutrients and vitamins) in the analyses.

We agree that this constitutes a limitation of the study. However, as stated in discussion section of the paper, our method is in line with that of the international recommendations for a varied and balanced diet with adequate intake of macro and micronutrients, which only consider nutrients derived from the diet (LINES 405-410).

  • It should be also more strongly underlined that over or underreporting of nutrients intake could occur when the data are collected by the questionnaire and that the declared food intake may also not correspond to the nutritional status measured by laboratory means (the concentration in blood)  as a result of different bioavailability of nutrients from different food products and individual differences in metabolism. It would be useful (if available) to look at the micronutrients/vitamins level in biological samples (looking at the previous papers from PHIME cohort Zn concentration was assessed in this study). This should be at least discussed in the discussion section of the paper.

We agree with the referee on the limitations of the FFQ which is why we refer to Willett’s publication for a description of its inherent weaknesses (Willett W, 2012)

The suggestion of looking at micronutrients/vitamins levels in biological samples is very interesting, however, the main objective of the Mediterranean prospective PHIME cohort study was to assess the association between low-level mercury exposure from food consumption during pregnancy and child neurodevelopment at 18 months of age. The analysis of other minerals and vitamins in biological samples was not included in the protocol. Only Zn and Se concentrations in maternal blood samples were analysed but only for two of the four participating countries (Italy and Croatia).

  • I suggest adding the additional section – “conclusion”

We added a conclusion section in the paper (LINES 430-438) by using some parts of the discussion section (LINES 425-428), as suggested.

The manuscript has been checked by a native English-speaking colleague.

Round 2

Reviewer 1 Report

The 19 authors made no substantial changes to the paper.

This paper describes dietary habits, energy, and nutrients intake in pregnant women from four countries (Croatia; Greece; Italy; Slovenia). The data were collected in the PHIME study, a newborn cohort study carried out in Italy between 2007 and 2010. In the end, the authors used data from this old database and compared the results from four states.

First of all, the authors should clarify that the data are more than 10 years old. The comparison between the 4 countries did not lead to significant results. 

Author Response

  • The 19 authors made no substantial changes to the paper.

We believe we have substantially modified the original version of the paper, as confirmed by the comment of Referee 2: “I accept all explanations. In my opinion, the manuscript has been significantly improved”.

  • This paper describes dietary habits, energy, and nutrients intake in pregnant women from four countries (Croatia; Greece; Italy; Slovenia). The data were collected in the PHIME study, a new-born cohort study carried out in Italy between 2007 and 2010. In the end, the authors used data from this old database and compared the results from four states. First of all, the authors should clarify that the data are more than 10 years old.

We modified the discussion section of the paper to better explain this aspect in the revised manuscript. In fact, in lines 301-306, we specify that the data are over 10 years old but that the results are still relevant and, for this reason, we decided to present the data in this paper: “While it is true that the data were collected over ten years ago, and that dietary habits in pregnancy may have changed since then, the main nutritional concerns that emerge from this assessment (i.e., macro- and micro-nutrients intake and dietary choices) continue to be relevant and reflect those which are still recognized as the most problematic aspects of nutrition in the general population (excessive SFA, soluble carbohydrates and sodium in-take, etc.) Indeed, the mean energy intakes of the women in our study, despite differing slightly between countries, were in line with those reported in Blumfield’s systematic review for the European region [Blumfield et al.  Nutr. Rev. 2012], and in other studies [Savard et al. Nutrients 2018].”

  • The comparison between the 4 countries did not lead to significant results. 

While it is true that the comparison between the 4 countries did not lead to significant results, this was not in the aim of the paper. In fact, as specified in lines 81-87 of the introduction section: “The main objective of the paper is to provide a description of dietary habits, energy and nutrients intake in pregnant women enrolled in the four countries of the Mediterranean PHIME cohort. The secondary objective of the paper is to evaluate the adherence of women enrolled in the cohort to the dietary recommendations proposed by EFSA. The present study allows to better understand eating patterns during pregnancy and to identify critical aspects that should be addressed by health care services and health professional, through effective interventions for the promotion of healthy eating habits.”

Reviewer 3 Report

Dear authors

I accept all explanations. In my opinion, the manuscript has been significantly improved.

Author Response

We are grateful to the Reviewer for the positive feedback